# Cultural Competence and Nursing Work Environment: Impact on Culturally Congruent Care in Portuguese Multicultural Healthcare Units

**DOI:** 10.3390/healthcare12232430

**Published:** 2024-12-03

**Authors:** Gisela Teixeira, Ricardo Picoito, Filomena Gaspar, Pedro Lucas

**Affiliations:** 1Nursing Research Innovation and Development Centre of Lisbon (CIDNUR), Nursing School of Lisbon, 1600-190 Lisboa, Portugalprlucas@esel.pt (P.L.); 2Centro de Inovação e Investigação Clínica e Núcleo de Investigação e Formação em Enfermagem da Unidade Local de Saúde Lisboa Ocidental, 1449-005 Lisboa, Portugal

**Keywords:** cultural competence, culturally congruent care, cultural diversity, nursing administration research, work environment

## Abstract

Background: Cultural competence is central to ensuring effective culturally congruent care to patients and fostering positive work environments, particularly in multicultural settings. Objective: This study aimed to analyse the relationship between cultural competence, the nursing work environment, and the delivery of culturally congruent care in multicultural units of a healthcare organisation in Portugal. Method: This was a quantitative, descriptive, and cross-sectional study, targeting nurses from multicultural units. Data were collected using both online and paper-based questionnaires, which included the Cultural Competence Questionnaire for Help Professionals, the Nursing Work Index—Revised Scale (NWI-R-PT), and a single question assessing nurses’ perceptions of the adequacy of the culturally congruent care they provide. Results: A moderate, positive correlation was identified between cultural competence and the Fundamentals for Nursing, while the nursing work environment was influenced by organisational support, professional category, and unit type. Discussion: The findings suggest that enhancing cultural knowledge and technical skills and reinforcing management support may positively impact culturally congruent care delivery in multicultural settings. Conclusion: This study contributes to nursing knowledge by highlighting the complex interactions between cultural competence and the work environment in Portuguese multicultural healthcare units. Future research should explore the impact of transcultural nursing leadership on multicultural work environments and in the delivery of culturally congruent care.

## 1. Introduction

In 2023, there was an increase of 33.6% in the resident foreign population in Portugal compared to 2022, totalling more than one million foreign citizens holding a Residence Permit [1]. Given the rising cultural diversity in Portuguese healthcare settings, where there has been a significant increase in the immigrant population, it is crucial to understand how cultural competence and the nursing work environment impact the delivery of culturally congruent care. Immigrants in Portugal often face barriers to healthcare access, including discrimination; distrust towards health professionals; differing beliefs, religious practices, and cultural traditions; a lack of knowledge of how the Portuguese healthcare system works; a lack of interpreters; and linguistic barriers [2,3]. Portugal’s public healthcare service, while providing universal coverage, can be challenging for immigrants to navigate, especially for those unfamiliar with local administrative processes. Waiting times for treatment, the limited availability of culturally adapted services, and complex bureaucratic procedures can discourage immigrants from seeking care [2,3]. Additionally, the absence of intercultural mediators in most healthcare institutions, who have been identified as crucial in improving communication, addressing cultural differences, and ensuring equitable health outcomes by facilitating understanding and reducing systemic barriers [4], contributes to the persistence of language barriers, making it difficult for immigrants to fully understand treatment options or communicate their health concerns effectively.

Therefore, creating a culturally competent work environment that supports nurses in providing care tailored to patients’ diverse backgrounds is essential.

Cultural competence has been described as a central tool for promoting positive work environments within organisations [5,6] and planning and delivering culturally congruent care to patients [7]. Cultural competence is defined as a process of integrating knowledge, attitudes, values, beliefs, behaviours, skills, practices, and cross-cultural encounters that include effective communication and the delivery of high-quality, safe, accessible, evidence-based, and effective nursing care to individuals, families, groups, and communities from both similar and diverse cultural backgrounds [8]. It is a lifelong learning process in which one progresses from being unconsciously incompetent to becoming unconsciously competent, treating and respecting each person with dignity, compassion, care, and empathy [9]. This learning can result from frequent engagement with diverse groups, expanding knowledge and critical awareness, and providing opportunities for the reflection and analysis of professional performance, leading to a better understanding and capacity to appropriately serve all individuals who may appear, think, and behave differently [10].

As a process without an endpoint [11], cultural competence can be continuously improved over time through the development of the affective, cognitive, and psychomotor learning domains, which are essential for healthcare professionals to enhance health outcomes and address disparities among different ethnic groups [9]. Cultural awareness, cultural knowledge, and cultural skills are three central dimensions for the development of cultural competence, common across most conceptual models and measurement instruments related to cultural competence [11]. Nonetheless, the weight of organisational support has also been reported as a key dimension for healthcare professionals to develop their cultural competence [10,12,13].

A culturally competent work environment should be adopted within the organisational culture, as it promotes the comfort of its nurses and promotes the delivery of culturally congruent care [14,15]. Culturally congruent care is defined as the use of sensitive, creative, and meaningful care practices that align with the values, beliefs, and lifestyles of patients, providing them with beneficial and satisfactory healthcare, or assisting them in difficult life situations, disabilities, or death [16].

The nursing work environment has been associated with a wide range of outcomes for both patients and nurses [17]. The nursing work environment is defined by Lake [18] as the set of organisational characteristics within the work context that either facilitate or constrain nurses’ professional practice. Favourable nursing work environments contribute to higher levels of job satisfaction, retention, and reduced burnout among nurses, as well as improvements in the quality of patient care, safety, hospital care experience, and patient satisfaction with nursing communication [17,19,20,21,22,23].

Planning and implementing strategies to improve nurses’ work environment can become particularly challenging when there is cultural diversity within the nursing work environment. Culturally diverse teams bring a range of perspectives, experiences, and knowledge that can foster creativity in the workplace [24], and are a valuable resource for understanding the languages, cultures, values, and health beliefs of diverse patient populations—enhancing culturally competent, patient-centred care that meets the needs and expectations of immigrant and multicultural groups [3,24,25,26,27]. However, culturally diverse teams are also associated with increased conflicts, mistrust, higher turnover rates, and reduced organisational commitment and alignment with shared values [28]. In Portuguese nursing work environments, these challenges may disrupt team cohesion and hinder effective collaboration, ultimately impacting the quality of patient care. While challenges related to cultural diversity in healthcare workplaces are not unique to Portugal, there are notable contrasts when compared to other culturally diverse countries, such as the United States of America. With its long history of multiculturalism among nurses and patients, the United States has implemented extensive measures to address diversity challenges in healthcare, such as mandated language services in federally funded hospitals and comprehensive cultural competence training programmes for healthcare providers [29]. These initiatives underscore the importance of cultural competence in improving healthcare quality.

Despite the lack of studies demonstrating its impact on patient outcomes [30], the cultural competence of healthcare professionals has been associated with the quality of care provided, as well as with patients’ health, safety, satisfaction, trust, and adherence and compliance with treatments, as well as a reduction in health disparities [11,31,32,33,34]. According to Campinha-Bacote [7], there is a direct relationship between the level of cultural competence of healthcare professionals and their ability to provide culturally congruent services to patients. No studies have been found that explore the impact of the nursing work environment on nurses’ cultural competence, or vice versa. Additionally, there is no research on how these two factors influence the delivery of culturally congruent care to patients from diverse cultural backgrounds in Portugal. Therefore, this study aimed to analyse the relationship between cultural competence, the nursing work environment, and the delivery of culturally congruent care in multicultural units of a healthcare organisation in Portugal. Specifically, our research addresses the following questions:

(1) How does nurses’ cultural competence correlate with the nursing work environment?

(2) To what extent does cultural competence influence the provision of culturally congruent care?

(3) To what extent does the nursing work environment influence the provision of culturally congruent care?

By exploring these questions, this study provides valuable insights into the interplay between cultural competence, the nursing work environment, and the delivery of culturally congruent care, with a particular focus on Portuguese healthcare settings. The findings aim to benefit nurses working in multicultural healthcare environments in Portugal by highlighting essential factors that support both culturally congruent care and a positive work environment. To our knowledge, this is the first study conducted in Portugal to analyse these relationships.

## 2. Materials and Methods

### 2.1. Study Design

This study used a quantitative, descriptive, cross-sectional design to examine the relationships between cultural competence, the nursing work environment, and the delivery of culturally congruent care. This design allowed us to analyse correlations between key variables and to explore factors influencing culturally congruent care within a multicultural healthcare setting.

### 2.2. Participants

To accomplish the purpose of this study, units from two hospitals within Unidade Local de Saúde (ULS) of Lisboa Ocidental were selected by convenience with the assistance of two liaison nurses, one from each hospital. Participants included nurses from units with culturally diverse nursing teams, characterised by at least nationality-based diversity. Additionally, these units had experience providing care to patients from diverse cultural backgrounds. Each liaison nurse contacted the units within their hospital to determine whether they met these criteria, and units meeting these criteria were included in the study. All nurses from these culturally diverse units were eligible to participate, regardless of their cultural background or nationality. Nurses from units that did not meet these characteristics were excluded.

A power analysis was not conducted to determine the sample size, as a convenience sampling approach was used. This approach was selected due to the specific inclusion criteria, which required units with unique cultural diversity and experience serving multicultural patient populations. Given these criteria, the focus was on obtaining a representative sample of these particular units rather than achieving a statistically calculated sample size.

The ULS of Lisboa Ocidental comprises three hospitals and 19 healthcare centres in the western region of Lisbon. It strives to be recognised by the local community and the general public as an innovative institution that delivers humanised, high-quality, and timely healthcare. The ULS is committed to promoting areas of differentiation and excellence while ensuring social and environmental sustainability, as well as valuing its professionals. Its core values include a focus on the individuals seeking its services, humanisation, and non-discrimination [35].

### 2.3. Data Collection

Data were collected through an online questionnaire between February and June 2024, which was sent only to the units who met the inclusion criteria. A paper version was also provided in the units for those unable to access the questionnaire online. The questionnaire included the following sections:

A. Sociodemographic and professional data.

B. Cultural Competence Questionnaire for Help Professionals (QCC-PA). Developed by Suarez-Balcazar et al. [10] and adapted and validated for the Portuguese population by Gonçalves and Matos [13], the QCC-PA consists of 16 Likert-type items (1—Strongly Disagree to 6—Strongly Agree) and assesses four dimensions: Cultural Awareness (appreciation and understanding of other cultures, recognition of one’s own biases towards other cultures, and critical examination of privileged societal privileges); Cultural Knowledge (familiarity with the characteristics, history, values, beliefs, and behaviours of other cultures); Technical Skills (ability to adapt professional practice to meet the needs of multicultural populations); and Organisational Support (implementation of individual and organisational practices that enhance professionals’ ability to intervene in a culturally appropriate manner). In the adaptation and validation study of the QCC-PA for Portugal [13], the instrument demonstrated an internal consistency of α = 0.88. In the current study’s sample, the internal consistency was α = 0.76.

C. Nursing Work Index—Revised Scale (NWI-R-PT). Originally developed by Aiken and Patrician [36] and adapted and validated for the Portuguese context by Anunciada et al. [37], this instrument evaluates the nursing work environment across six dimensions: Management Support, Professional Development, Fundamentals for Nursing, Nurse–Physician Relationship, Endowments, and Organisation of Nursing Care. The NWI-R-PT consists of 31 Likert-type items, with responses ranging from 1—Strongly Disagree to 5—Strongly Agree. The NWI-R-PT demonstrated excellent internal consistency (α = 0.91), maintaining a similar level of reliability in our study (α = 0.92).

D. A single question on nurses’ perception of the adequacy of their culturally congruent care delivery. This Likert-type question ranged from 1—Absolutely Inappropriate to 5—Absolutely Appropriate in asking nurses to evaluate the adequacy of the culturally congruent care they provide to patients from diverse cultural backgrounds.

### 2.4. Data Analysis

A comprehensive analysis of the study variables was conducted, including descriptive, comparative, and correlational analyses. Descriptive statistics were used to summarise the demographic and professional characteristics of the sample, while comparative analyses examined differences between groups. Correlational analyses were conducted to explore the relationships between key variables. To ensure an appropriate analysis for each variable, normality assumptions were assessed using the Shapiro–Wilk test Q-Q plots. Additionally, the homogeneity of variance was evaluated using Levene’s test. When either normality or homogeneity of variance assumptions were violated, non-parametric tests were used, such as the Mann–Whitney U test or Kruskal–Wallis test, in place of parametric tests, in line with the recommendations of Marôco [38].

Multiple linear regression models were estimated to assess the influence of predictor variables on the nursing work environment and on nurses’ cultural competence. Sociodemographic and professional variables were included as potential confounders in the models to control for their effects.

Missing data were handled using pairwise deletion. This approach was chosen to maximise the use of available data and to reduce the impact of missing responses on the overall findings.

All tests were performed at a 5% significance level using IBM SPSS Statistics Version 27.

### 2.5. Ethical Considerations

Data collection began after receiving approval from the Ethics Committee of ULS Lisboa Ocidental (approval code 2407). Participant anonymity and confidentiality were guaranteed. The study was fully explained in the informed consent form, included in both the online and paper versions of the questionnaire, and data collection only proceeded with participants’ voluntary consent.

## 3. Results

### 3.1. Sample Characteristics

A total of 155 nurses from 10 units participated in this study. Over 80% were female, 15.5% were male, and 0.6% were non-binary, with ages ranging from 33 to 69 years (M = 39.1; SD = 11.2). About 4% of the nurses were non-Portuguese, specifically from Brazil (1.3%) and Spain (2.6%). The majority of participants identified as Catholic (74.2%), with 10.3% identifying as Atheist, 5.8% as Agnostic, 1.3% as Orthodox, 0.6% as Adventist, and 7.1% as belonging to other religions.

The sample included nurse managers (6.5%), nurse specialists (20.6%), and staff nurses (72.9%). The average length of professional experience was 16.2 years (SD = 11.4), ranging from less than 1 year to 46 years. Participants worked in Surgical (23.2%), Labor and Delivery (17.4%), Emergency (11.6%), Orthopaedics (11%), Obstetrics (8.4%), Outpatient Clinic (7.7%), Contingency/Bed Reserve (7.1%), Oncology Day Hospital (6.5%), Neuro Trauma (3.9%), and Neurology (3.2%) units.

Of the 155 nurses, 83.2% reported not having training in multiculturalism and 15.5% did not provide a response to this question. About 7% reported having worked in other countries, namely, in Brazil, Spain, Guinea-Bissau, and the United Kingdom.

In total, 40% of nurses providing direct patient care (nurse specialists and staff nurses) rated their delivery of culturally congruent care to patients from diverse cultural backgrounds as appropriate, 25.2% as somewhat appropriate, 18.1% as absolutely appropriate, and 5.8% as inappropriate. Seventeen nurses did not respond to the single question regarding the adequacy of their culturally congruent care delivery.

### 3.2. Cultural Competence

The mean score for *Cultural Competence* was 3.32 (SD = 0.61), with the highest mean score in the *Cultural Knowledge* dimension (M = 3.97; SD = 1.05) and the lowest in the *Technical Skills* dimension (M = 2.77; SD = 0.84). Table 1 presents the mean scores and standard deviations for all dimensions of Cultural Competence.

No significant differences were found in the mean score for *Cultural Competence* across different genders (*H*(2) = 1.177; *p =* 0.555), nationalities (*H*(2) = 2.357; *p =* 0.308), religions (*H*(5) = 4.419; *p =* 0.491), academic qualifications (*H*(3) = 2.887; *p =* 0.409), or units (*H*(9) = 11.135; *p =* 0.267). Similarly, no significant differences were found in the mean score for *Cultural Competence* across professional categories (nurse manager, nurse specialist, and staff nurse) (*H*(2) = 1.960; *p* = 0.375). Additionally, no significant differences were observed in the mean scores for *Cultural Competence* between nurses who had received or were currently undergoing multicultural training and those who had not (*U* = 566.000; *p =* 0.543), nor between those who had worked abroad and those who had never worked outside of Portugal (*U* = 31.500; *p =* 0.561). Additionally, no significant correlations were found between *Cultural Competence* and age (*r* = 0.044; *p =* 0.615) or length of professional experience (*r* = 0.033; *p =* 0.703).

### 3.3. Nursing Work Environment

The mean score for *Nursing Work Environment* was 3.30 (SD = 0.51), with the highest scores observed in the dimensions of *Nurse–Physician Relationship* (M = 3.69; SD = 0.67), *Fundamentals for Nursing* (M = 3.60; SD = 0.52), and *Management Support* (M = 3.60; SD = 0.74). The lowest mean scores were found in the dimensions of *Professional Development* (M = 2.73; SD = 0.72) and *Endowments* (M = 2.60; SD = 0.85). Table 2 presents the mean scores and standard deviations for *Nursing Work Environment* and its dimensions.

Statistically significant differences in *Nursing Work Environment* were observed across units (*H*(9) = 33.082, *p* < 0.001). The highest mean score was found in the Contingency/Bed Reserve unit (M = 3.97; SD = 0.38), which was significantly different from the Outpatient Clinic (*p* < 0.001), Neurology (*p* = 0.001), Labor and Delivery (*p* < 0.001), Emergency (*p* < 0.001), Oncology Day Hospital (*p* = 0.006), Orthopaedics (*p* = 0.013), Surgery (*p* = 0.005), and Neuro Trauma (*p* = 0.039) units. The Outpatient Clinic and Neurology units had the lowest mean scores, with no significant differences between them (*p* = 0.947).

### 3.4. Correlations Between Cultural Competence and the Nursing Work Environment

A moderate and significant positive correlation was found between *Cultural Competence* and *Fundamentals for Nursing* (*r* = 0.250; *p* = 0.003). Additionally, moderate and significant positive correlations were observed between *Organisational Support* and the overall *Nursing Work Environment* (*r* = 0.338; *p* < 0.001), as well as with *Management Support* (*r* = 0.270; *p* = 0.001), *Professional Development* (*r* = 0.297; *p* < 0.001), and *Fundamentals for Nursing* (*r* = 0.290; *p* < 0.001). Table 3 summarises the correlations between *Cultural Competence* and *Nursing Work Environment* and their respective dimensions.

### 3.5. Impact of Cultural Competence in Multicultural Nursing Work Environments

A multiple linear regression with stepwise variable selection was employed to estimate a parsimonious model for predicting the *Nursing Work Environment* in multicultural healthcare units. The independent variables included *Cultural Competence* and its dimensions (*Cultural Awareness*, *Cultural Knowledge*, *Technical Skills*, and *Organisational Support*), as well as sociodemographic and professional variables such as gender, age, academic qualifications, professional category, length of professional experience, and unit.

The assumptions of the final model—normal distribution, homogeneity, and independence of errors—were tested. Only normality of the residuals was not confirmed (KS(137) = 0.087; *p* = 0.014). Homogeneity was validated with the White test (*X*^2^(123) = 126.057; *p* = 0.407), and the assumption of independence was confirmed using the Durbin–Watson statistic (d = 1.80), indicating no autocorrelation between residuals, as described by Marôco [38]. The VIF was used to diagnose multicollinearity, and no issues were detected (VIF < 5).

The stepwise multiple linear regression analysis revealed that the professional category (β = 0.203; *p* = 0.009), unit (β = 0.265; *p* < 0.001), and *Organisational Support* (β = 0.314; *p* < 0.001) were significant predictors of the *Nursing Work Environment* (Table 4). The final adjusted model was significant, explaining 21.4% of the variability of the *Nursing Work Environment* in the multicultural units of ULS Lisboa Ocidental (F(3, 133) = 13.320; *p* < 0.001; $R_{a}^{2}$ = 0.214).

The positive coefficient of 0.202 indicates that for each additional unit increase in the mean score of *Organisational Support*, there is an expected 0.202-unit increase in the mean score of *Nursing Work Environment*, assuming all other variables remain constant.

Compared to Contingency/Bed Reserve Unit (unit of reference), the mean score of *Nursing Work Environment* was lower by 0.790 units in the Labor and Delivery unit (*p* < 0.001), lower by 0.931 units in the Outpatient Clinic (*p* < 0.001), lower by 0.603 units in Surgery (*p* < 0.001), lower by 1.129 units in Neurology (*p* < 0.001), lower by 0.599 units in Obstetrics (*p* = 0.002), lower by 0.690 units in the Oncology Day Hospital (*p* = 0.001), lower by 0.681 units in the Emergency unit (*p* < 0.001), lower by 0.611 units in Orthopaedics (*p* < 0.001), and lower by 0.590 units in Neuro Trauma (*p* = 0.017). Additionally, the mean score for *Nursing Work Environment* was lower by 0.179 units among Nurse Specialists compared to Staff Nurses, although this was not significant (*p =* 0.084).

### 3.6. Impact of Multicultural Nursing Work Environments on Cultural Competence

Likewise, a multiple linear regression with stepwise variable selection was performed to estimate a parsimonious model for predicting *Cultural Competence*, based on the *Nursing Work Environment* and its dimensions (*Management Support*, *Professional Development*, *Fundamentals for Nursing*, *Nurse–Physician Relationship*, *Endowments*, and *Organisation of Nursing Care*), as well as gender, age, nationality, religion, academic qualifications, professional category, length of professional experience, experience working abroad, and multicultural training. The stepwise multiple linear regression analysis identified the *Fundamentals for Nursing* (β = 0.287; *p* = 0.007) as the only significant predictor of nurses’ *Cultural Competence* in the multicultural units of the healthcare organisation under study (Table 5).

Although the final model was statistically significant, it explained only 5.4% of the variability of nurses’ *Cultural Competence* (F(1, 112) = 7.441; *p* = 0.007; $R_{a}^{2}$ = 0.054), suggesting that most of the variability is influenced by factors not included in this study.

The positive coefficient of 0.287 indicates that for each additional unit increase in the mean score of *Fundamentals for Nursing*, there is an expected increase of 0.287 units in the mean score of nurses’ *Cultural Competence*. Both normality of the residuals (KS(138) = 0.062; *p* = 0.200) and homoscedasticity (*X*^2^(72) = 71.330; *p* = 0.500) were confirmed. Independence of the residuals was confirmed by the Durbin–Watson statistic (d = 2.14), and no multicollinearity issues were detected (VIF < 5).

### 3.7. Association Between Cultural Competence and the Delivery of Culturally Congruent Care

Significant differences were found in the mean scores of Cultural Knowledge (H(3) = 8.817; *p* = 0.032) between nurses with different levels of culturally congruent care delivery. Multiple comparison analysis revealed that those differences occurred between nurses who rated their care as “Inappropriate” and “Absolutely Appropriate” (*p* = 0.018), as well as between “Somewhat Appropriate” and “Absolutely Appropriate” (*p* = 0.014). As observed in Figure 1, the score of *Cultural Knowledge* was lower among nurses who considered their care “Inappropriate” (M = 3.42; SD = 0.58) and higher among those who considered their care “Absolutely Appropriate” (M = 4.38; SD = 1.03).

Significant differences were also observed in the mean scores for *Technical Skills* (*H*(3) = 9.993; *p* = 0.019). The multiple comparison analysis indicated differences between nurses who rated their care as “Absolutely Appropriate” and “Appropriate” (*p* = 0.034), and between “Absolutely Appropriate” and “Somewhat Appropriate” (*p* = 0.002). The mean score for *Technical Skills* was lower among nurses who rated their care as “Absolutely Appropriate” ($\overset{-}{x}$ = 2.35; SD = 0.93) and higher among those who rated it as “Somewhat Appropriate” ($\overset{-}{x}$ = 3.01; SD = 0.75) (Figure 2).

No significant differences were found for *Cultural Competence*, *Cultural Awareness*, and *Organisational Support* between nurses with different perceptions of culturally congruent care delivery (*p* > 0.05). However, it was noticed that the mean score of overall *Cultural Competence* was higher among nurses who rated their delivery of culturally congruent care as “Absolutely Appropriate” (M = 3.36; SD = 0.63) and lower among those who rated it as “Inappropriate” (M = 3.03; SD = 0.37).

### 3.8. Association Between the Nursing Work Environment and the Delivery of Culturally Congruent Care

Significant differences were found in the mean score for *Management Support* between nurses with different levels of culturally congruent care delivery (F = 3.153; *p* = 0.027). A multiple comparison analysis revealed that these differences occurred between nurses who rated their culturally congruent care as “Inappropriate” and “Somewhat Appropriate” (*p* = 0.039), “Inappropriate” and “Appropriate” (*p* = 0.015), and “Inappropriate” and “Absolutely Appropriate” (*p* = 0.038). The lowest mean score for *Management Support* was observed among nurses who considered their culturally congruent care as “Inappropriate” (M = 2.89; SD = 0.61) (Figure 3).

No significant differences were found between nurses with different perceptions of culturally congruent care delivery regarding the *Nursing Work Environment* and the remaining dimensions (*p* > 0.05). Nevertheless, it was noticed that the mean score for *Nursing Work Environment* was higher among nurses who rated their delivery of culturally congruent care as “Appropriate” (M = 3.38; SD = 0.49) and lower among those who rated it as “Inappropriate” (M = 3.00; SD = 0.45).

## 4. Discussion

The present study is the first to explore the relationships between Cultural Competence, the Nursing Work Environment, and the delivery of culturally congruent care in multicultural units of a Portuguese healthcare organisation. Its findings contribute to the nursing knowledge on cultural competence and its impact on nursing work environments and the care provided to patients from diverse cultural backgrounds, highlighting several areas for improvement in training, support, and workplace dynamics.

One of the significant findings was the limited number of nurses with multicultural training, with only 1.3% (N = 2) of participants having received such training. This highlights a pressing need for healthcare organisations to integrate cultural competence training into professional development programmes. Training in multiculturalism, particularly in settings with diverse patient populations, is essential for equipping nurses with the skills to deliver culturally congruent care and work effectively in multicultural teams. The more the organisational environment formally and informally supports and encourages the assessment and delivery of culturally congruent services, the more likely healthcare professionals are to develop cultural competence [10,12]. Organisations should provide opportunities for cultural competence certification [39], as such training equips professionals to work effectively and deliver culturally appropriate healthcare to patients from diverse cultures and those speaking different languages [40].

Our analysis found no significant associations between cultural competence and sociodemographic or professional variables. Similar to the study by Tolentino Diaz et al. [41], our study did not find significant associations between nurses’ cultural competence and participation in multicultural training programmes. Contrary to other studies, no significant relationships were found between nurses’ cultural competence and variables such as academic qualifications, length of professional experience [41,42,43], age, or nationality [44]. This suggests that cultural competence, as measured in this study, may be an intrinsic attribute rather than one influenced by demographic factors. However, discrepancies with other studies could be attributed to variations in sample sizes or cultural competence assessment tools, indicating a need for further research on the factors influencing cultural competence.

The Nursing Work Environment score in our study was slightly higher than in some previous studies [45,46] and slightly lower than in others [47,48], suggesting variability in nursing work environments across different settings, whether they are culturally diverse or not. However, most of these studies utilised the Practice Environment Scale of the Nursing Work Index (PES-NWI) developed by Lake [18], which employs a four-point Likert scale. In contrast, our study used a five-point Likert scale to assess the nursing work environment, which limits the comparability of the findings.

The Contingency/Bed Reserve unit, in particular, showed a significantly higher mean score compared to other units. This unit operates differently from the rest—it is a transitional service, activated based on the needs of the ULS. As such, nurses float from other units, resulting in a unique work environment. Initially, no one knows each other, and a shared sense of mission unites the team in their efforts.

The high scores in the dimensions of Management Support, Nurse–Physician Relationships, and Fundamentals for Nursing suggest that the role of nurse manager, the collaboration between healthcare professionals, and the adherence to core nursing principles are strengths in multicultural healthcare units. On the other hand, the lower scores in Professional Development and Endowments point to potential areas for improvement. These results are similar to those found by Lucas et al. [22] in primary healthcare in Portugal. These findings are also consistent with international literature that emphasises the need for continuous professional development and adequate resources in multicultural settings [49,50,51].

The relationship between Cultural Competence and the Nursing Work Environment were of particularly interest. The findings suggest that enhancing Organisational Support and reinforcing the Fundamentals for Nursing may lead, respectively, to improvements in the work environment and to higher levels of cultural competence among nurses. The positive correlation of Organisational Support with several dimensions of the Nursing Work Environment suggests that improvements in institutional support for nurses to engage in culturally competent practices are associated with a more positive work environment, and underscores the need for healthcare institutions to prioritise culturally competent practices as part of their organisational policies. According to Balcazar et al. [12], the extent to which an organisation supports its professionals’ engagement in culturally competent practices is crucial. Nurses, in particular, need to develop their cultural competence, and healthcare organisations should provide the necessary support for this development [52].

The multiple linear regression analysis identified the organisational support, professional category, and unit as significant predictors of the Nursing Work Environment, emphasising the importance of the institution’s implementation of individual and organisational practices that enhance nurses’ ability to intervene in a culturally appropriate manner in multicultural units. The fact that nurses working in multicultural units reported lower mean scores in the work environment compared to other units suggests that additional support and resources may be necessary to ensure that these units have more positive work environments, conducive to high-quality care delivery.

The impact of the Nursing Work Environment on Cultural Competence, as revealed by the multiple regression analysis, was primarily associated with the Fundamentals for Nursing. This finding may highlight the importance of units adopting conceptual frameworks, theoretical models, and nursing theories in transcultural nursing. Transcultural nursing involves a theoretical and evidence-based approach that focuses on culturally based beliefs, values, and practices related to health and illness [8]. By integrating transcultural nursing principles, such as reflective self-assessment, patient-centred communication, and a holistic understanding of cultural diversity [8], healthcare units can foster an environment that supports the development of cultural competence among nurses. Once the model explained only a small portion of the variability in nurses’ cultural competence, it points to the need for further research to explore additional factors that may influence nurses’ cultural competence in Portuguese multicultural settings.

The analysis of the relationship between Cultural Competence and the delivery of culturally congruent care provides valuable insights. Nurses who rated their care delivery as “Absolutely Appropriate” had significantly higher scores in Cultural Knowledge and Technical Skills compared to those who rated their care as “Inappropriate” or “Somewhat Appropriate.” This suggests that a greater familiarity with the characteristics, history, values, beliefs, and behaviours of other cultures, along with a strong capacity to adapt professional practice to meet the needs of multicultural populations, may lead to improved delivery of culturally congruent care. However, controlled trials and mixed-method studies are needed to determine whether Cultural Knowledge and Technical Skills genuinely enhance the delivery of culturally congruent care by measuring their impact on patient outcomes. According to a literature review by Vella et al. [53], some studies report patient perceptions of improved cultural competence among healthcare professionals following training. Nonetheless, none of the studies included in the review demonstrated clinically significant improvements in patient health outcomes.

Last of all, our findings suggest that higher levels of Management Support are associated with better perceptions of culturally congruent care delivery, emphasising the potential influence of nursing management in promoting effective care for diverse populations. This result may start filling the void of inexistent research exploring the delivery of culturally congruent care as an outcome enhanced by nursing management and leadership in multicultural healthcare units [54]. As suggested by the findings of Dauvrin and Lorant [55], the level of cultural competence among leaders seems to partially contribute to the cultural competence level of healthcare professionals, which, in turn, appears to have a positive impact on patients. Once it is believed that nurse managers’ cultural competence and transcultural leadership may play a crucial role in promoting culturally congruent care delivery to patients from diverse cultural backgrounds [56,57], future studies should also explore these hypothetic relationships.

## 5. Limitations and Recommendations

Despite the valuable insights provided by this study, some limitations must be acknowledged. The small sample size and the predominantly Portuguese population may limit the generalisability of the findings to broader, more diverse healthcare settings. This demographic profile may influence certain aspects of cultural competence and the nursing work environment, potentially impacting the applicability of the results to other cultural contexts. Future studies with larger and more diverse samples are recommended to enhance the applicability of these findings across different cultural contexts.

The cross-sectional design restricts the ability to establish causal relationships between variables, meaning that any correlations observed should not be interpreted as causal. Additionally, the study relied on psychological scales to measure complex constructs such as cultural competence and the nursing work environment. While these scales provide useful quantitative insights, they may not capture the full depth of these constructs. Treating these scales as purely numerical data can be limiting, as changes in scores may have nuanced implications for nursing practice that are not fully conveyed by statistical results alone.

The low explanatory power of the regression models suggests that other factors, not examined in this study, may play a significant role in shaping both cultural competence and the nursing practice environment. Future research should explore additional variables that may influence cultural competence, such as personal attitudes, prior intercultural experiences, and broader organisational policies that support cultural competence. Studies examining how transcultural leadership impacts the nursing work environment, nurses’ cultural competence, and the delivery of culturally congruent care would also be valuable. Conducting these studies with larger and more diverse samples across various healthcare settings could improve the generalisability of the findings.

Finally, given that a notable percentage of nurses rated their delivery of culturally congruent care as “Somewhat Appropriate” (25.2%) and “Inappropriate” (5.8%), we recommend the development and validation of an instrument for the Portuguese context to accurately assess the delivery of culturally congruent nursing care and help healthcare organisations identify areas for improvement and implement strategies to enhance care quality.

## 6. Conclusions

This study highlights the complex relationship between cultural competence, nursing work environments, and the delivery of culturally congruent care in multicultural units. Our findings suggest that cultural competence and a supportive nursing work environment are mutually reinforcing, with higher levels of cultural knowledge, technical skills, and management support contributing to improved culturally congruent care. To enhance care quality for diverse patient populations, healthcare organisations should consider strategies that bolster both cultural competence and work environment quality, such as increasing organisational support for culturally competent practices and reinforcing nursing fundamentals.

As cultural diversity continues to grow in Portuguese healthcare organisations, these findings highlight the need for further research into the specific factors influencing cultural competence and work environments. Future studies should aim to develop targeted interventions that empower nurses to deliver culturally congruent care, thereby improving health outcomes in increasingly multicultural settings and minimising the barriers that immigrants in Portugal face when accessing healthcare services.

## Figures and Tables

**Figure 1 healthcare-12-02430-f001:**
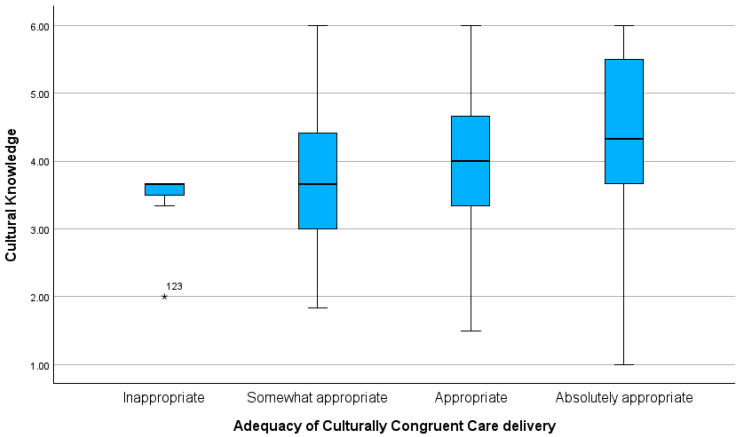
Comparison of Cultural Knowledge scores across nurses’ perceptions of the adequacy of culturally congruent care delivery.

**Figure 2 healthcare-12-02430-f002:**
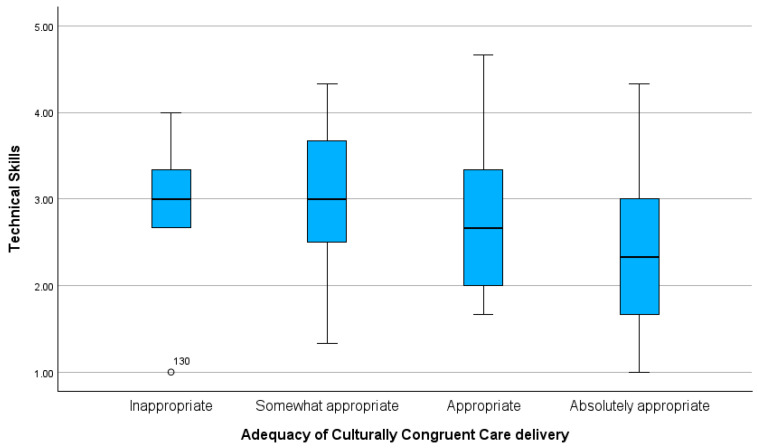
Comparison of Technical Skills scores across nurses’ perceptions of the adequacy of culturally congruent care delivery.

**Figure 3 healthcare-12-02430-f003:**
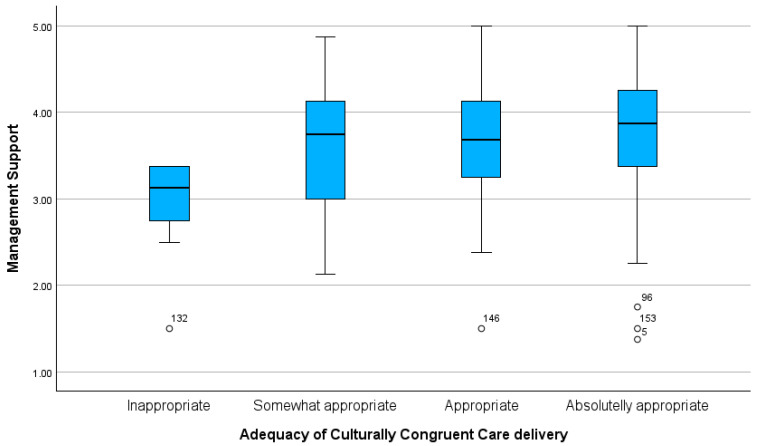
Comparison of Management Support scores across nurses’ perceptions of the adequacy of culturally congruent care delivery.

**Table 1 healthcare-12-02430-t001:** Mean and standard deviation of Cultural Competence and its dimensions.

Dimensions	Mean	SD
Cultural Competence	3.32	0.61
Cultural Awareness	3.05	0.72
Cultural Knowledge	3.97	1.05
Technical Skills	2.77	0.84
Organisational Support	2.93	0.79

**Table 2 healthcare-12-02430-t002:** Mean score and standard deviation of Nursing Work Environment and its dimensions.

Dimensions	Mean	SD
Nursing Work Environment	3.30	0.51
Management Support	3.60	0.74
Professional Development	2.73	0.72
Fundamentals for Nursing	3.60	0.52
Nurse–Physician Relationship	3.69	0.67
Endowments	2.60	0.85
Organisation of Nursing Care	3.26	0.65

**Table 3 healthcare-12-02430-t003:** Correlations between Cultural Competence and Nursing Work Environment.

	Cultural Competence	Cultural Awareness	Cultural Knowledge	Technical Skills	Organisational Support
**Nursing Work Environment**	0.220 **	0.102	0.131	−0.028	0.338 **
Management Support	0.186 *	0.089	0.118	−0.009	0.270 **
Professional Development	0.077	0.044	−0.037	−0.055	0.297 **
Fundamentals for Nursing	0.250 **	0.114	0.234 **	−0.103	0.290 **
Nurse–Physician Relationship	0.225 **	0.147	0.142	0.082	0.223 **
Endowments	0.170 *	0.042	0.243	0.053	0.202 *
Organisation of Nursing Care	0.098	0.026	0.058	−0.060	0.194 *

** Correlation is significant at 0.01 level; * Correlation is significant at 0.05 level.

**Table 4 healthcare-12-02430-t004:** Multiple linear regression analysis predicting the Nursing Work Environment.

Model	Unstandardised Coefficients		Standardised Coefficients	t	*p*-Value	95% Confidence Interval for B	
	B	Std. Error	Beta			Lower Limit	Upper Limit
(Constant)	2.003	0.242		8.294	<0.001	1.525	2.481
Organisational Support	0.202	0.050	0.314	4.057	<0.001	0.104	0.301
Unit	0.026	0.008	0.265	3.436	<0.001	0.011	0.041
Professional category	0.175	0.066	0.203	2.656	0.009	0.045	0.305
Gender			0.015	0.193	0.847		
Age			−0.113	−1.313	0.191		
Academic qualifications			0.003	0.028	0.977		
Length professional experience			−0.069	−0.790	0.431		
Cultural Awareness			−0.016	−0.194	0.846		
Cultural Knowledge			0.060	0.693	0.490		
Technical Skills			−0.104	−1.356	0.177		
Overall Cultural Competence			0.019	0.166	0.868		

**Table 5 healthcare-12-02430-t005:** Multiple linear regression analysis predicting Cultural Competence.

Model	Unstandardised Coefficients		Standardised Coefficients	t	*p*-Value	95% Confidence Interval for B	
	B	Std. Error	Beta			Lower Limit	Upper Limit
(Constant)	2.262	0.383		5.905	<0.001	1.503	3.021
Fundamentals for Nursing	0.287	0.105	0.250	2.728	0.007	0.079	0.496
Gender			−0.033	−0.356	0.723		
Age			0.105	1.124	0.264		
Nationality			−0.011	−0.120	0.905		
Religion			0.035	0.379	0.705		
Academic qualifications			0.094	1.024	0.308		
Professional category			0.002	0.021	0.983		
Worked Abroad			−0.066	−0.717	0.475		
Multicultural Training			−0.046	−0.496	0.621		
Length of Professional Experience			0.089	0.948	0.345		
Management Support			0.084	0.804	0.423		
Professional Development			−0.064	−0.601	0.549		
Nurse–Physician Relationship			0.137	1.321	0.189		
Endowments			0.104	1.090	0.278		
Organisation of Nursing Care			−0.012	−0.116	0.908		
Overall Nursing Work Environment			0.085	0.643	0.521		

## Data Availability

The original contributions presented in the study are included in the article; further enquiries can be directed to the corresponding author.

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
