# Peer review of "Cultural Competence and Nursing Work Environment: Impact on Culturally Congruent Care in Portuguese Multicultural Healthcare Units"

_healthcare, 2024, doi:10.3390/healthcare12232430_

Round 1

Reviewer 1 Report

Comments and Suggestions for Authors

The subject matter of the manuscript is indeed compelling; however, the current state of the manuscript necessitates extensive revisions to address significant shortcomings and to meet the standards of rigorous research.

Introduction

Lines 3645

Please provide a detailed overview of the healthcare access barriers faced by immigrants in Portugal, supported by existing research.

Lines 8199

Please provide a detailed overview of the healthcare access barriers faced by immigrants in Portugal, supported by existing research.

Please outline the impact of the Portuguese healthcare system on immigrants' ability to access healthcare.

Using relevant literature, please elaborate on the negative consequences of cultural diversity within Portuguese nursing workplaces.

Please compare and contrast the challenges faced by countries with cultural diversity, such as other European nations and the United States, to those faced in Portugal.

While analysis is a method, please clarify the specific research questions you aim to answer through your analysis. Furthermore, how will the findings of your study benefit nurses working in Portuguese healthcare settings? Please provide a clear and concise explanation of your study's purpose and significance.

2.2. Participants

Lines 105117

Please provide a more detailed explanation of how the liaison nurses selected the study participants. Specifically, what criteria were used to identify units with cultural diversity? 

Was a power analysis conducted to determine the appropriate sample size for this study? If not, please justify the decision to not perform a power calculation.

Please provide supporting literature to substantiate your description of the Unidade Local de Saúde (ULS) units.

2.3. Data collection

A detailed description of the sample's characteristics, such as age, gender, and profession, is essential for understanding the findings.

It seems that data for item [C] is missing. Please double-check.

How many units were surveyed, and what were their characteristics?

2.3. Data analysis

The data analysis methods should be described in more detail. Please provide a clear and comprehensive explanation of how each variable was handled, ensuring that the analysis approach is appropriate for the research questions and measurement levels of the variables. For instance, you mentioned using non-parametric tests when normality assumptions were violated, but how did you determine that normality was not met? A detailed description of the data analysis methods is crucial for assessing the reproducibility and reliability of the study.

Results

There are two sections labeled 3.2. Please double-check for any errors before submission.

Overall, the results section is poorly organized and difficult to follow. The main research questions are unclear. When presenting data in tables, the text should focus on the most important findings and avoid redundancy. Minimizing overlap between tables and text can reduce reader fatigue and emphasize key results.

There is no need to present all variables' mean and standard deviation. Focus on the most important variables related to your research questions. 

Given that this study relies heavily on psychological scales, it is important to acknowledge the limitations of treating these scales as purely numerical. When interpreting changes in scale scores, consider the practical implications for nursing practice. Additionally, be cautious not to confuse correlation with causation.

Given these comments, a complete overhaul of the results section is strongly recommended to enhance clarity and coherence.

Discussion

The discussion section lacks coherence and is overly lengthy. The discussion should prioritize the interpretation of results relevant to the research objectives, avoiding unnecessary details. Similar to the results section, the discussion should be focused on the variables relevant to the study's objectives and should be written in a clear and concise manner.

The significant disconnect between the analysis, results, and discussion is evident. The discussion frequently veers off into a literature review rather than providing a deep dive into the implications of the current study's findings. The focus should be on interpreting the primary outcomes in the context of the study's objectives.

When interpreting the results, it is recommended to consider the characteristics of the sample. In this case, given that the sample is primarily Portuguese, this fact should be reflected in the discussion.

As mentioned earlier, the limitations section should include important limitations such as the small sample size, the predominantly Portuguese population, and the treatment of psychological measures as numerical variables.

Conclusion

Based on these points, I strongly recommend a significant revision of the conclusion section.

Author Response

Dear reviewer,

Thank you for your wise comments to our manuscript. Kindly find the attachment with the responses and improvements done to the manuscript.

Best regards,

The authors

Reviewer 2 Report

Comments and Suggestions for Authors

Dear Author,

Thank you for your effort in studying such an important topic among nurses.

I have some suggestions:

·       Please add some information about sociodemographic and professional data.

·       Please add sample size calculation

·       Please add inclusion and exclusion criteria

Good luck.

Author Response

Dear reviewer,

Thank you for your comments to our manuscript. Kindly find the attachment with the responses and improvements done to the manuscript.

Best regards,

The authors

Reviewer 3 Report

Comments and Suggestions for Authors

Dear Editor, 

Thank you for the opportunity to evaluate the manuscript healthcare-3287296, entitled “Cultural Competence and Nursing Work Environment: Impact on Culturally Congruent Care in Portuguese Multicultural Healthcare Units”. According to the authors, “Background: Cultural competence is central to ensuring effective culturally congruent care to patients and to foster positive work environments, particularly in multicultural settings. Objective: This study aimed to analyze the relationship between cultural competence, the nursing work environment, and the delivery of culturally congruent care in multicultural units of a healthcare organization in Portugal. Method: This was a quantitative, descriptive, and cross-sectional study, targeting nurses from multicultural units. Data were collected using both online and paper-based questionnaires, which included the Cultural Competence Questionnaire for Help Professionals, the Nursing Work Index-Revised Scale (NWI-R-PT), and a single question assessing nurses' perceptions of the adequacy of the culturally congruent care they provide. Results: A moderate, positive correlation was identified between cultural competence and the Fundamentals for Nursing, while the nursing work environment was influenced by organizational support, professional category, and unit type. Discussion: The findings suggest that enhancing cultural knowledge, technical skills, and reinforcing management support can improve the delivery of culturally congruent care. Conclusion: This study contributes to nursing knowledge on cultural competence in Portuguese multicultural healthcare units and highlights the importance of leadership in fostering culturally competent nursing work environments. Future research should investigate the impact of transcultural nursing leadership on multicultural work environments and in the delivery of culturally congruent care”.

The topic is relevant to public health and in a brief survey, it was not possible to identify similar texts online, which guarantees the potential for citation in a possible publication. I would like to point out that the manuscript has weaknesses that can be addressed to improve the text. For the analysis, I used the STROBE protocol (Strengthening the reporting of observational studies in Epidemiology).

The manuscript is very well written. It allows the reader to fully understand the topic. in the first lines of the summary. It presents the knowledge gaps in its introduction and identifies immigration not as a problem, but as a cultural challenge.

Furthermore, I indicate below questions from STROBE that need to be integrated into the text. Please note that many of these questions have already been addressed. are integrated into the text.

1. Indicate the study design in the title or abstract, using a commonly used term. 2. Provide an informative and balanced summary of what was done and what was found in the abstract. 3. Detail the theoretical framework and the reasons for conducting the research. 4. Describe the specific objectives, including any pre-existing hypotheses. 5. Present, at the beginning of the article, the key elements related to the study design. 6. Describe the context, relevant locations and dates, including the periods of recruitment, exposure, follow-up and data collection. 7. Cohort Studies: Present the eligibility criteria, sources and methods of participant selection. Describe the follow-up methods. Case-Control Studies: Present the eligibility criteria, sources and diagnostic criteria for identifying cases and the methods of selecting controls. Describe the rationale for selecting cases and controls. Cross-sectional study: Present the eligibility criteria, sources, and methods of participant selection. Cohort studies: For paired studies, present the pairing criteria and the number of exposed and unexposed individuals. Case-control studies: For paired studies, present the pairing criteria and the number of controls for each case.

8. Clearly define all outcomes, exposures, predictors, potential confounders, and effect modifiers. When necessary, present diagnostic criteria.

9. For each variable of interest, provide the source of the data and details of the methods used in the assessment (measurement). When there is more than one group, describe the comparability of the assessment methods.

10. Specify all measures adopted to avoid potential sources of bias.

11. Explain how the sample size was determined.

12. Explain how the quantitative variables were treated in the analysis. If applicable, describe the categorizations that were adopted and why.

13. Describe all statistical methods, including those used to control for confounding. Describe all methods used to examine subgroups and interactions.

14. Explain how missing data were handled. Cohort Studies: If applicable, explain how losses to follow-up were handled. Case-Control Studies: If applicable, explain how matching of cases and controls was handled. Cross-Sectional Studies: If applicable, describe the methods used to consider the sampling strategy. Describe any sensitivity analyses.

15. Describe the number of participants at each stage of the study (e.g., number of potentially eligible participants, screened according to eligibility criteria, actually eligible, enrolled in the study, completed follow-up, and actually analyzed). Describe the reasons for losses at each stage. Consider the relevance of presenting a flow diagram. 16. Describe participant characteristics (e.g., demographic, clinical, and social) and information on exposures and potential confounders. Indicate the number of participants with missing data for each variable of interest. Cohort Studies: Present the follow-up period (e.g., mean and total time).

17. Cohort Studies: Describe the number of outcome events or summary measures over time. Case-Control Studies: Describe the number of individuals in each exposure category or present summary measures of exposure. Cross-Sectional Studies: Describe the number of outcome events or present summary measures.

18. Describe the unadjusted estimates and, if applicable, the estimates adjusted for confounding variables, as well as their precision (e.g., confidence intervals). Make clear which confounders were used in the adjustment and why they were included. When continuous variables are categorized, report the cutoff points used. If appropriate, consider transforming the relative risk estimates into absolute risk for a relevant time period.

19. Describe other analyses that were performed, e.g., subgroup analyses, interaction analyses, sensitivity analyses.

20. Summarize the main findings by relating them to the study objectives.

21. Present the limitations of the study, taking into account potential sources of bias or imprecision. Discuss the magnitude and direction of potential biases.

22. Present a cautious interpretation of the results, considering the objectives, limitations, multiplicity of analyses, results from similar studies, and other relevant evidence.

23. Discuss the generalizability (external validity) of the results.

24. Specify the source of funding for the study and the role of the funders. If applicable, provide such information for the original study on which the article is based.

- Results and discussion

The results were well described and presented categories of analysis relevant to the research. Discussion is important and uses classic literature on the subject, as well as recent texts. I suggest incorporating international sources.

Recommendation: Minor revisions.

Author Response

(The authors gave the same response as above.)

Round 2

Reviewer 1 Report

Comments and Suggestions for Authors

Thank you for your careful revision of the manuscript.

Please provide citations for the revised sentences below: 

Lines 45−53

Portugal's public healthcare service, while providing universal coverage, can be challenging for immigrants to navigate, especially for those unfamiliar with local administrative processes. Long waiting times, limited availability of culturally adapted services, and complex bureaucratic procedures can discourage immigrants from seeking care. Additionally, the absence of standardized protocols for interpreter services means that language barriers persist, making it difficult for immigrants to fully understand treatment options or communicate their health concerns effectively, thereby impacting the quality and accessibility of care.

The content of lines 138-146 describes the research methodology. As this information belongs in the 'Methods' section, it should either be deleted from its current location or relocated to the appropriate section. If relocating, please ensure that there is no redundancy and that the content is reorganized for clarity.

Author Response

Dear reviewer,

Thank you for your revision and comments for improvement. Kindly find the attachment with the our responses.

Warm regards
